# Evidence-Based Lessons from Policy Implementation Research in Two Countries Achieving Progress on Global Breastfeeding Targets: Recommendations from the Philippines and Viet Nam

**DOI:** 10.3390/healthcare13050544

**Published:** 2025-03-03

**Authors:** Catherine Pereira-Kotze, Paul Zambrano, Tuan T. Nguyen, Janice Datu-Sanguyo, Duong Vu, Constance Ching, Jennifer Cashin, Roger Mathisen

**Affiliations:** 1Alive & Thrive, Global Nutrition, FHI 360, Washington, DC 20037, USA; jennifer.cashin1@gmail.com; 2Alive & Thrive, Global Nutrition, FHI 360, Quezon City 1101, Philippines; pzambrano@fhi360.org; 3Alive & Thrive, Global Nutrition, FHI 360, Hanoi 11022, Vietnam; tnguyen@fhi360.org (T.T.N.); vduong@fhi360.org (D.V.); cching@fhi360.org (C.C.); rmathisen@fhi360.org (R.M.); 4Alive & Thrive, Global Nutrition, FHI 360, Muntinlupa 1770, Philippines; jcdatu@gmail.com

**Keywords:** breastfeeding, maternal, infant, and young child nutrition, policy implementation, International Code, maternity protection, maternal health, care economy, gender equality, the Philippines, Viet Nam

## Abstract

**Background/Objectives**: There is extensive evidence that breastfeeding saves lives, improves health, and provides value to the economy and societies worldwide. The Philippines and Viet Nam have progressive policies to enable breastfeeding, and breastfeeding rates in these countries have substantively improved. In the Philippines, exclusive breastfeeding under six months (EBF) increased from 35.9% (2008) to 60.1% (2021) and, in Viet Nam, in just over a decade, EBF has more than doubled, from 17.0% (2010) to 45.4% (2021). We aimed to use an evidence synthesis to consolidate learnings from policy support to enable breastfeeding in the Philippines and Viet Nam, to identify insights to improve future programming to improve breastfeeding practices in these two countries, and glean learnings that can potentially be adapted for similar contexts. **Methods**: This manuscript presents a qualitative evidence synthesis of seven purposively selected research articles from the Philippines and Viet Nam. **Results**: Since the 1960s, the Philippines and Viet Nam have notably improved policies protecting breastfeeding. Both countries have implemented legislation that is substantially aligned with the International Code of Marketing of Breast-milk Substitutes and subsequent World Health Assembly resolutions. Both countries have improved paid maternity leave provisions, with Viet Nam providing 6 months of paid leave, yet insufficient coverage to informal workers, and the Philippines providing 3.5 months of paid maternity leave while expanding maternity protection coverage to informal workers. From 2006–2021, breastfeeding rates increased in both countries alongside policy improvements yet barriers to implementation remain. **Conclusions**: Implementation research has documented policy progress and improved breastfeeding rates in the Philippines and Viet Nam. Our analysis offers valuable lessons potentially applicable beyond these contexts, emphasizing the need for addressing policy gaps and targeted cross-sectoral policy actions to enhance breastfeeding practices. Learnings from implementing national marketing restrictions of commercial milk formula and associated products and maternity protection policies in these countries could inform the implementation of newly developed regional standards together with supportive policies, facilitating the harmonization of regional regulatory environments.

## 1. Introduction

There is conclusive evidence that breastfeeding saves lives, improves health, and supports economies in countries worldwide [1,2,3,4]. The actions required to improve breastfeeding rates to protect the health of women and children are well-established [1,2,5,6]. Yet even though progress in achieving breastfeeding goals has been made, improvements are unequal across countries and remain insufficient to meet the Global Nutrition Target that 50% of infants under six months of age be exclusively breastfed (EBF) by 2025 and the Global Breastfeeding Collective target of 70% EBF by 2030 [7,8]. Many global declarations, policy guidelines, and recommendations for protecting, promoting, and supporting breastfeeding have been made over the past 45 years [9,10,11]. Since 2017, the Global Breastfeeding Collective has prioritized seven policy actions for countries to protect and support breastfeeding [8]: funding, code implementation, maternity protection, the Baby-Friendly Hospital Initiative (BFHI), breastfeeding counseling and training, community support, monitoring, and infant and young child feeding (IYCF) support in emergencies. Restrictions on the marketing of commercial milk formula (CMF) and improved maternity protection are two priority actions that can protect breastfeeding.

The need to protect breastfeeding from harmful marketing of CMF has been emphasized since 1939 when the first public call for penalties on such marketing was made [12,13]. The International Code of Marketing of Breast-Milk Substitutes (hereafter, “the Code”) was first adopted by the World Health Assembly (WHA) in 1981 as a minimum standard, thereby mandating countries to implement corresponding national legal measures [9]. The text of the 1981 Code needs to be interpreted alongside all subsequent resolutions that reinforce and give clarity to its scope and coverage [14,15,16]. Adequate monitoring and enforcement are essential for successful implementation of such legislation. There is still wide variation in the adoption and implementation of the Code across countries. As of March 2024, 146 (75%) of 194 World Health Organization (WHO) member states had adopted legal measures implementing at least some provisions of the Code, while 48 countries still had no national Code legislation [17]. The WHO regions of Africa, the eastern Mediterranean, and Southeast Asia have the highest percentage of countries substantially aligned with the Code. Exploitative marketing, notably the evolving marketing tactics and political economy underpinning the CMF industry, remains a concern globally, posing significant barriers to breastfeeding [6,18,19].

The Philippines and Viet Nam are two Southeast Asian countries and member states of the Association of Southeast Asian Nations (ASEAN) with progressive policies to enable breastfeeding. The Philippines was one of the first countries to legislate the Code in 1986 and, in 2006, updated its implementation rules to substantially align with the Code [17]. The CMF industry in the Philippines is dominated by two international companies, Nestle and Reckitt Benckiser (Mead Johnson), controlling 94% of the market share in 2020 [20]. The use of corporate political activity by the CMF industry in the Philippines has been documented but was met with public health resistance [20]. The Vietnamese legislation has been moderately aligned with the Code since 2014 [17] and is one of few countries to incorporate restrictions on marketing CMF for pregnant or breastfeeding women even though manufacturers widely disregard these. The CMF industry in Viet Nam is competitive, has experienced recent growth, and includes a variety of domestic and global manufacturers [21]. In 2018, at least 28 new formula products entered the market [22]. In 2019, Vinamilk and Nutifood made up 36% of the Vietnamese CMF market, followed by Abbott (17%), Friesland Campina (11%), Mead Johnson (9%), Nestle (8%), and others (9%) [23]. The size of the CMF industry in the Southeast Asia region has expanded [24], and the region has been a target for increased CMF marketing and production [25,26].

The International Labour Organization (ILO) first defined maternity protection over 100 years ago [27]. The current Maternity Protection Convention (MPC) 183 and Recommendation 191 of 2000 define comprehensive maternity protection for working women who are pregnant, around the time of childbirth, or breastfeeding as including health protection at the workplace, a period of maternity leave with accompanying cash and medical entitlements, job security and income protection, non-discrimination, breastfeeding or expressing breaks, and access to childcare support [28,29]. Governments must invest in comprehensive maternity protection, as it serves as a form of social security that safeguards women workers. This protection ensures their income, employment, and health—along with the wellbeing of their infants—are not jeopardized by pregnancy, childbirth, breastfeeding, or maternity leave [30]. In addition to being a labor rights issue, maternity protection is now acknowledged as a public health intervention to support breastfeeding with importance to society, for sustainable development and gender equity, and fulfilling women’s reproductive rights during the periods of pregnancy, childbirth, and sexed infant care work such as breastfeeding [31,32]. Extending paid maternity leave has positive implications for breastfeeding practices [33,34], maternal, infant, and young child health, and maternal mental health [35,36,37]. Similarly, women’s return to work is an established barrier to EBF or breastfeeding continuation [38]. Globally, women carry a disproportionate burden (more than 75%) of unpaid care work [39]. Access to comprehensive maternity protection could contribute to reducing this inequality. Despite the establishment and existence of global standards for maternity protection, as of February 2025, only 44 countries (and not a single country in Asia) have ratified the ILO MPC 183 [40], even though many countries meet several minimum standards of the convention [41]. Even when maternity protection is legislated, access and implementation challenges remain, especially for vulnerable groups such as “non-standard” workers who usually have minimal and inconsistent access to informal support [42,43,44].

Despite not yet ratifying ILO MPC 183 [40], the Philippines has multiple policies for maternity protection, including provisions for paid lactation breaks, establishment of breastfeeding spaces, protection against gender discrimination, and other forms of workplace breastfeeding support. It was not until 2019 that the duration of paid maternity leave was extended from 8–11 weeks/60–78 days to 15 weeks/105 days [45] to align with ILO standards. The female labor force participation rate in the Philippines is 52.9% compared to males at 76.3% [46]. However, rates of informal employment are not routinely reported and have recently been redefined by the ILO. It has recently been estimated that overall informal employment in the Philippines (men and women combined) may exceed 80% [47]. The Government of Viet Nam has advanced gender equality policies and was an early adopter of maternity protection, currently providing 26 weeks (6 months) of paid maternity leave to women working formally, even without having ratified the ILO MPC 183. Viet Nam also has a higher female labor force participation rate (61.6% at 2021) than regional and global averages, even though female labor force participation has decreased slightly since 1990 [48]. There is a high rate of informality among female workers in Viet Nam (67.2% in 2021) [49], while rates of vulnerable employment among women have decreased from 86.4% in 1991 to 57.3% in 2022 [48].

The Philippines was one of the first countries in the world [50] and Viet Nam was the first country in Asia and second in the world to ratify the Convention on the Rights of the Child (CRC). This is relevant because Article 18 of the CRC describes State Parties’ obligations and parents’ responsibilities relating to child rearing and Article 24 describes governments’ obligations to respect, protect, and fulfill children’s rights to the highest attainable standard of health and nutritious foods, including women’s rights to have skilled support to enable breastfeeding. The CRC has acknowledged that the public must be protected from improper and biased information, and therefore, implementation of the Code is necessary for governments to fulfill their obligations under the CRC [51,52,53,54].

The Philippines and Viet Nam have made progress in legislating the Code and maternity protection, implementing the Baby-Friendly Hospital Initiative, early essential newborn care practice standards, breastfeeding counseling as part of essential health services, and, in Viet Nam, designated Centers of Excellence for Breastfeeding [55]. Notwithstanding fundamental differences in governance structures in the Philippines and Viet Nam and growing commercial influence, policies have remained protective of breastfeeding, likely due to technical assistance and concerted advocacy support from multiple stakeholder countries [56,57,58,59,60,61]. Breastfeeding rates in these two countries have subsequently increased in recent years and this enabling policy environment therefore warrants further in-depth investigation and documentation. This is because, despite extensive evidence on the importance of breastfeeding and established global policies to support it, there are still significant gaps in understanding the factors that hinder the effective implementation of these policies across diverse settings. Research on the specific challenges countries face in enforcing the Code and comprehensive maternity protection legislation, especially in regions with low adherence, is limited. Additionally, there is a need for more in-depth studies on the role of informal work and economic disparities in limiting access to breastfeeding support and on how targeted advocacy can create enabling environments for breastfeeding in low-resource settings. In this study, we synthesized findings from published research coordinated by the Alive & Thrive initiative (A&T) and documents on policy implementation in the Philippines and Viet Nam. The aim of this evidence synthesis is to identify insights to improve future programming, including identifying gaps requiring further action to improve breastfeeding practices in these two countries and consolidate lessons learned from policy support to enable breastfeeding that can potentially be adapted for similar contexts.

## 2. Materials and Methods

### 2.1. Study Design

This manuscript presents a qualitative evidence synthesis of seven purposively selected primary research publications from the Philippines and Viet Nam, summarized in Table 1. The primary research selected for this evidence synthesis was coordinated by the (A&T and based on the terms of funding to work with previously collected data or information of this initiative. A&T is a global nutrition initiative aiming to strengthen nutrition programming to subsequently improve the health and wellbeing of women, children, and adolescents. A&T does this through high-quality, large-scale social and behavior change communication, explicit focus on policy and advocacy to achieve and sustain change, promoting country-led initiatives through technical assistance, and prioritizing quality assurance and improved programming [62]. The original studies were selected because they were conducted in the Philippines and Viet Nam, were coordinated by A&T, and their scope included the Code and/or maternity protection (i.e., strategies to protect breastfeeding). The seven published studies selected used mixed methods: in-depth interviews, cross-sectional surveys, and secondary data analysis following a published research protocol [57].

### 2.2. Study Setting

The Philippines is a lower-middle-income country (LMIC) that is a democratic constitutional republic with a highly decentralized government. Table 2 provides further sociodemographic indicators for the Philippines. The proportion of infants EBF has increased from 29.7% in 2003 to 60.1% in 2021 [70,71] (Figure 1). The Philippines has, therefore, met the Global Nutrition Target of 50% by 2025, and if this rate of increase is maintained, it will meet the target of 70% by 2030. Despite this encouraging trend, it is important to acknowledge that the Philippines has large geographic inequalities in EBF practices [72]. Viet Nam is a lower-middle-income country that has a socialist republic with a highly centralized government. Table 2 provides further sociodemographic information for Viet Nam. Rates of EBF under six months increased in Viet Nam from 17.0% in 2010 to 24.0% in 2014 and 45.4% in 2021 [73,74], demonstrating progress towards achieving the global nutrition targets of 50% by 2025 and 70% by 2030 [75] (Figure 1). Similarly to the Philippines, even within the context of increased national EBF rates, Viet Nam also has geographic inequalities in EBF practices [72]. Some breastfeeding indicators (early initiation of breastfeeding and bottle feeding) and the rates of cesarean sections worsened between 2011 and 2020 [76]. Per capita expenditure on CMF in Viet Nam is almost double that of the Philippines (Table 2). This could be due to higher rates of exclusive and continued breastfeeding in the Philippines compared to Viet Nam and/or unnecessary supplementation with CMF in children over one year of age, when they could be fed with complementary foods and regular animal milks (Table 2).

### 2.3. Collating, Synthesizing, and Reporting the Results

All the studies included were read and summarized by CPK. Data from the seven included studies were extracted and synthesized using deductive content analysis to code and categorize information. The conceptual model proposed in the protocol was used as a framework to describe how breastfeeding rates increased in the context of evolving structural and environmental determinants of breastfeeding, including strengthened regulation of CMF marketing and improved maternity protection [57]. A more detailed description of the seven studies is provided in Table 1. The various syntheses that were conducted are presented in the results as boxes, tables, and figures, including milestones of main changes to legislation, lessons learned, barriers to implementing legislation, and recommendations to improve policy implementation to enable breastfeeding in the Philippines and Viet Nam.

### 2.4. Ethical Considerations

Since this manuscript presents a synthesis of secondary data, ethical approval was not required for this review. For the primary data collection of the seven articles included in this review, appropriate ethical approvals were obtained, and ethical principles were used in the primary data collection. The original studies were conducted according to the guidelines of the Declaration of Helsinki and approved by the Institutional Review Board (or Ethics Committee) of FHI 360 (protocol code 1383644; approved on 16 April 2019), the Hanoi University of Public Health (protocol code 019-501/DD-YTCC; approved on 12 June 2019), and St. Cabrini Medical Center—Asian Eye Institute Ethics Review Committee (SCMC-AEI ERC) (protocol code 2020-027; approved on 20 November 2020).

## 3. Results

### 3.1. Overview of Policies to Enable Breastfeeding in the Philippines and Viet Nam

There are comprehensive and long-standing policies and legislation to protect, promote, and support breastfeeding in the Philippines and Viet Nam, many of which are consistent with global standards. Figure 2 summarizes the key policies and legislation to protect breastfeeding (i.e., key milestones for maternity protection provision and Code implementation). Box 1 and Table 3 summarize Code implementation legislation in the Philippines and Viet Nam. Table 4 and Figure 2 summarize paid maternity leave provision in the Philippines and Viet Nam, compared to global guidance.

Box 1Summary of legislation implementing the Code in the Philippines and Viet Nam.Philippines (“Substantially aligned” with the Code) 1986 Executive Order No. 51, National Code of Marketing of Breastmilk Substitutes, Breastmilk Supplement and Other Related Products (also known as “The Philippines Milk Code” of 1986) [88].2006 Administrative Order No. 2006–0012, Revised Implementing Rules and Regulations (RIRR) of 2006 [89].2012 Joint Administrative Order No. 2012-0027 (The Inter-Agency Committee (IAC) Guidelines in the Exercise of the Powers and Functions as stated in EO 51) [90].Viet Nam (“Substantially aligned” with the Code)2012 Law on Advertising (No. 16/2012/QH13) bans advertising of breastmilk substitutes up to 24 months of age, complementary foods for children < 6 months, feeding bottles, and teats [91].2014 Decree on trading in & using of nutritional products for infants, feeding bottles & teats (No. 100-2014-ND-CP) [92].2020 Decree on sanctioning of administrative violations in health sector (No. 117/2020/ND-CP) [93].2021 Decree on sanctioning of administrative violations in advertising (No. 38/2021/ND-CP) [94].2022 Law on inspection (No. 11/2022/QH15) [95].

### 3.2. Implementation of Code Legislation in the Philippines and Viet Nam

The Philippines and Viet Nam have implemented legislation that aligns with the Code for over 20 years. The Philippines has been substantially aligned with the Code since 1986 [17,67]. The establishment of an Inter-Agency Committee (IAC) overseeing all advertising and promotional materials within the scope of the Philippine Milk Code [67] has been described as the best-functioning part of the monitoring and enforcement system in the Philippines [96]. In Viet Nam, Decree 100 was moderately aligned with the Code in 2014, but since 2020, legislation has been substantially aligned. Viet Nam is one of the only countries with some, albeit limited, restrictions on the marketing of CMF for pregnant women (CMF-PW). Real-time policy evaluations from 2015 to 2017 in Viet Nam identified opportunities used to strengthen national Code legislation. These include integrating Code provisions into existing laws, namely the Advertisement Law. This allowed a more simplified process by already having a legal basis for developing subsequent sub-laws and preventing the need to create new enforcement mechanisms from the start [63]. Another opportunity in Viet Nam was the leveraging of advocacy efforts to maintain stringent advertising bans. In mid-2016, when companies opened the vote and were advocating for the advertising ban to be lowered to 12 months (from 24 months), one champion acted quickly to counter this, making a compelling case to voting members of Parliament by framing the extension of the advertisement ban to 24 months as a clear and unambiguous child rights and people-centered approach. This was successful in ensuring that the scope of the advertising ban was maintained and demonstrated the need for constant vigilance against industry tactics [63]. In Viet Nam, during this process, letters were sent to companies describing forthcoming legislative changes and their obligations and the government prioritized training workshops, dissemination of compliance results, and media monitoring, leading to successful enforcement [63]. In Viet Nam, “street-level bureaucrats”, described as public sector workers detecting and reporting violations of legislation, were reported as playing an important role in Code implementation, and it would therefore be strategic to ensure they are adequately trained [60]. The achievements in Viet Nam have served as a model for progress in other countries in the region.

However, reported Code violations and implementation of legislation in Viet Nam show inconsistent compliance. Data from 2020 indicate a reduction in the promotion of CMF by health workers in health facilities, correlating with high compliance scores on the latest Code Status Report [16,64]. Hospital and health leaders demonstrate a good understanding of the Code, supported by policies prohibiting company representatives from promoting substitutes at facilities. The Ministry of Health (MoH) oversees the monitoring of Decree 100 through various agencies, including the Viet Nam Food Administration (VFA) and health inspection bodies. Code implementation in health facilities is integrated into routine assessments as one of 83 criteria of the National Hospital Standards and Accreditation for both public and private hospitals, with self-assessments followed by verification from MoH and provincial health departments [64,96]. Outside the health system, the VFA conducts periodic inspections of food safety and labeling, while promotional activities at the point of sale are monitored by the Ministry of Commerce’s Inspectors [96]. The Philippines and Viet Nam are two of the thirty-seven countries that have explicitly included provisions regulating the promotion of breastmilk substitutes on digital platforms in their current national legislative frameworks.

### 3.3. Implementation of Maternity Protection Legislation in the Philippines and Viet Nam

Both countries meet the minimum requirements of the ILO MPC 183 for the length of paid maternity leave provided, the maternity leave cash payments at 100% of previous earnings and provision of paid breastfeeding breaks [41]. Maternity protection entitlements in the Philippines have recently been improved to align more closely with international standards [69] with the Expanded Maternity Leave Law (EMLL) increasing paid maternity leave to 15 weeks in 2019, in line with the ILO MPC 183 but not the ILO MP Recommendation 191 (18 weeks) nor WHO EBF recommendation (26 weeks) (Figure 2). In the Philippines, there has been mixed source funding for cash payments, from social security and employers, since 2019. The Labor Code of the Philippines and EMLL prohibit discrimination against employing women [45,97]. Medical entitlements for mothers are provided through the Philippine Health Insurance Corporation (PhilHealth), a government-managed social health insurance program. There are comprehensive policies to mandate workplace breastfeeding support [69], for example, the Expanded Breastfeeding Promotion Act of 2009 mandates [98]:establishment of lactation rooms;implementation of lactation breaks in workplaces;workplaces are required to create a breastfeeding policy;workplaces are required to comply with the Philippine Milk Code;compliance with the Act is required to issue/renew business permits;workplaces can apply for renewable exemptions in establishing lactation rooms if exemptible criteria are metworkplaces can apply for Mother–Baby-Friendly Workplace Certification (valid for two years) by complying with this Act and fulfilling additional requirements set by the Department of Health (DoH).○Review and assessment of applications are assigned to local government units.○Onsite inspection and approval of certification are conducted by DoH Centers for Health Development.

The implementation of these lactation support policies was reported to vary according to workplace and type of work. Some employers reported having additional policies such as flexible working arrangements, free transport, additional cash assistance, free medical check-ups, and/or free childcare [69].

Viet Nam has advanced maternity protection policies, providing six months of maternity leave paid at 100% of previous earnings to women employed formally, with high uptake and moderate awareness of maternity protection entitlements among formally employed women [66]. In Viet Nam, 91.7% of formally employed women contribute to the public social insurance fund, Viet Nam Social Security (VSS), coordinated by the Ministry of Labour, Invalids and Social Affairs [66].

### 3.4. Barriers to Implementing Legislation to Enable Breastfeeding in the Philippines and Viet Nam

Even after more than two decades of national legislation implementing the Code in the Philippines and Viet Nam, there are still implementation gaps in both countries, with key barriers relating to adoption, enactment, monitoring, and enforcement of legislation (Table 5).

### 3.5. Recommendations to Improve the Implementation of Policies to Enable Breastfeeding in the Philippines and Viet Nam

The articles reviewed and synthesized provided comprehensive recommendations for how the implementation of policies to enable breastfeeding in the Philippines and Viet Nam could be achieved, summarized in Table 6. Both countries should implement social and behavior change communication strategies to raise awareness of Code legislation and promote breastfeeding. Improved breastfeeding counseling, community and health system support, and continuous training for health professionals are essential. Effective monitoring, political commitment, and enforcement are crucial to protect breastfeeding (Table 6). To improve maternity protection in the Philippines and Viet Nam, paid maternity leave should be extended to six months in the Philippines. It should be more accessible for the informal sector in both countries by reducing contribution requirements and expanding coverage. Strengthening communication about entitlements, promoting supportive work environments, and improving awareness of social security programs are crucial. Strengthened policy enforcement, inter-agency monitoring, and public or social insurance funding for paid leave are needed. Government–workplace partnerships should support breastfeeding programs.

## 4. Discussion

Through a qualitative evidence synthesis of purposefully selected studies, we have provided an in-depth analysis of policy support to enable breastfeeding in the Philippines and Viet Nam. Through consolidating the results and recommendations from primary research conducted in the two countries, we can provide specific recommendations for policy improvements relating to the restriction of marketing of breastmilk substitutes and associated products and maternity protection policies. These include addressing any remaining legislative gaps, implementing accompanying social and behavior change communication strategies, capacity building of stakeholders with responsibility for policy implementation, integrated monitoring and enforcement, improved coverage and uptake of policies, sustainable financing mechanisms, strengthened collaboration between government and partners, and policy alignment across sectors. Lessons learned from the in-depth investigation in these two countries could potentially be adapted for similar contexts, depending on characteristics such as knowledge, attitudes, practices, and contextual barriers or enablers relating to marketing restrictions and maternity protection policies.

The positive policy environment in the Philippines and Viet Nam has likely played an important role in contributing to increased EBF rates in both countries over the past two decades. However, gaps remain in regulatory frameworks that serve to protect breastfeeding. Some laws still lack clarity, and the full implementation and impact of recent legislation are still emerging. Recent studies indicate that, while progress has been made, there is still room for better alignment between national nutrition strategies and global standards in Southeast Asia [100,101]. Additionally, the lack of policy coherence and disjointed links with efforts to address individual barriers through social and behavior change communication continue to limit policy impact and effectiveness.

### 4.1. National Code Legislation in the Philippines and Viet Nam

Given the size and power of the CMF industry in the region, regulation (through national Code and maternity protection legislation) becomes a necessary strategy to counteract this dominance and ensure a fairer playing field for public health. Gaps in the content of Code legislation in the Philippines and Viet Nam are clear. Both countries need to determine actions to fully implement recent WHO guidance on regulatory measures to restrict digital marketing of breastmilk substitutes [102]. The CMF industry’s “marketing playbook” has been well-described and documented [18], including the use of cross-promotion and product range extension beyond common CMF products. Another example of this is recent research from Viet Nam reporting that 19.5% of mothers had given commercially produced colostrum milk powder to infants within the first six months of life [103]. Policymakers should ensure these evolving marketing strategies are acknowledged when strengthening national Code legislation. Deciding on the most appropriate mechanism to improve Code legislation is complex, with risks and benefits. However, the actors involved should ensure that industry lobbying does not weaken existing legislation. There are examples of successful resistance to corporate political activity from both the Philippines and Viet Nam [20,58] and governments need to recognize evolving tactics to indirectly weaken national Code legislation, such as lobbying around corporate social responsibility [104].

The results from this evidence synthesis demonstrate how manufacturers exploit any gaps in national legislation for marketing purposes, with persistent Code violations in both countries despite strong legislation [64,67,105,106,107]. The Code and subsequent WHA resolutions should be viewed as minimum standards, which countries should adopt in full, aiming to score 100/100 in Code Status Reports, together with robust monitoring and enforcement. This is essential in the context of pervasive marketing. The Philippines and Viet Nam experience challenges similar to those faced by many countries in monitoring and enforcing Code legislation. The NetCode Ongoing Monitoring System Protocol should be institutionalized in all countries, with strong civil society involvement [99]. Monitoring and enforcement should align with new WHO digital marketing recommendations, potentially leveraging artificial intelligence (AI) tools to support monitoring activities with support from civil society organizations to enhance sustainability. In the Philippines, a citizen reporting platform was developed that encouraged crowd-based monitoring [108,109,110]. Unfortunately, there were challenges in maintaining enforcement actions against violators of the Philippines Milk Code [96], and these efforts were not sustained beyond 2019. In Viet Nam, the Virtual Violations Detector (VIVID) was launched in October 2022 and developed through a partnership between A&T and the Vietnamese government. VIVID is an automated online program using AI and supervised machine learning to detect violations of Code-relevant national legislation on digital platforms [111,112,113]. This tool has successfully detected large numbers of online Code violations [114], demonstrating the potential for sustainable government-led monitoring to ensure legislation enforcement on digital platforms.

There may be many factors influencing parents and caregivers’ infant feeding practices, and these influences may have different impacts on breastfeeding initiation and continuation in different regions of the world. However, it is important that choices relating to infant feeding are made in an enabling environment that is free from commercial influence. The value of the Code is that it aims to protect all infants from harmful marketing, whether they are fed with breastmilk or infant formula.

### 4.2. Maternity Protection Legislation in the Philippines and Viet Nam

In the Philippines, the results highlight the need for improved social and behavior change communication policies [69], which is crucial to raising awareness of their importance for enabling breastfeeding and advancing policy coherence. The challenges in accessing maternity protection entitlements in the Philippines, despite their availability, are similar to the barriers in accessing cash payments during maternity leave observed in South Africa [42,115]. Complex administrative processes and bottlenecks hinder women, particularly informal workers, from fully realizing their entitlements. The ILO has recently recommended removing administrative barriers to accessing social insurance, especially among informal workers [116]. Governments should consider implementing measures such as Centers of Excellence for Breastfeeding and context-specific toolkits to support breastfeeding in the workplace, as Viet Nam [117,118] and the Philippines have done [119].

The conflicting finding in Viet Nam that improved paid maternity leave policies do not necessarily increase the likelihood of women having a paid, formal job 3–5 years after giving birth [68] reveals potentially negative implications for women and demonstrates the unintended consequences of policy incoherence. Reasons for limitations on women’s labor force participation despite policy reforms include maternity leave pay being less than previous earnings, late payment of maternity entitlements, and inaccessible, unaffordable childcare for children under two years [68]. A contrasting positive implication of improved maternity protection policy in Viet Nam is that longer paid maternity leave is associated with increased formal employment and less unpaid work among women [120]. Therefore, it is important for all policies to incorporate provisions that protect and promote women and children’s rights. Integrating other international labor conventions can help strengthen and improve maternity protection policies [60]. Women working informally or in rural areas still have limited access to maternity protection in Viet Nam [60], prompting advocacy for policy expansion [121]. Viet Nam’s Social Insurance Law is currently being amended to provide a one-time maternity allowance to informal sector employees who contributed to the voluntary insurance scheme, but the proposed amount is minimal (USD 90). The allowance should be increased to at least the level of the urban poverty line [43] to ensure that mothers and babies do not enter into poverty due to childbirth, and entitlements should be expanded to all informal workers, with adequate budgetary support [121]. The cost for providing adequate maternity allowances is less than the health consequences for mothers and children when paid maternity leave and breastfeeding support are unavailable or inaccessible [121]. A key lesson from Viet Nam’s maternity protection policy is that inclusive maternity protection for formal and informal workers should not be separate advocacy exercises, and all working women should be eligible for maternity protection, regardless of their place or sector of work.

International conventions do not yet stipulate paternity leave but the Philippines and Viet Nam provide four and five days of paid paternity leave, respectively [41]. Without appropriate paternity leave provisions, women continue to bear the disproportionate load of unpaid domestic and care work. These unequal gender norms perpetuate gender inequality. The governments of the Philippines and Viet Nam should ratify ILO MPC 183 to ensure accountability for providing maternity protection entitlements. ILO conventions are not universally ratified, and in LMICs, this may be influenced by the economic costs of ratification [122]. Countries should further be encouraged to implement maternity protection policies that go beyond ILO minimum requirements. Global organizations have started to recommend six months of paid maternity leave at 100% pay to all women, regardless of income, employment, or immigration status [32].

A recent systematic literature review reported that over 80% of studies described the impact of returning to work on breastfeeding duration and CMF initiation as key aspects of women’s experiences when returning to work following maternity leave [38]. Work-related factors that facilitate breastfeeding continuation upon return to work include workplace breastfeeding support policies, lactation room facilities, breaks for expressing breastmilk, breastfeeding support groups, on-site childcare [38], and paid maternity leave. This review recommended more research on women’s lived experiences when returning to work after maternity leave and especially the impact on maternal mental and physical health.

### 4.3. Cross-Sectoral Nature of Breastfeeding Policy Implementation

In both the Philippines and Viet Nam, cross-cutting barriers have been documented that impact the implementation of policies related to the Code and maternity protection. In the Philippines, there are individual barriers (knowledge and skills gaps among mothers, fathers, and health workers; infant feeding misconceptions; low confidence among mothers partly due to insufficient support) and structural barriers (inconsistent breastfeeding promotion in communities, workplaces, and households), with many women describing the return to work as a reason for using CMF despite awareness of the immune protection conferred by breastfeeding [67]. In Viet Nam, gaps in health systems practices interrupt breastfeeding policy implementation. Almost 90% of women in Viet Nam bring CMF with them or purchase CMF at the facility when they give birth, presumably because they are worried about breastmilk supply in the first few days [64,76]. This corresponds to high CMF use, with 79.3% of mothers providing CMF in the first three days [76]. Unsurprisingly, this corresponds to a low prevalence of early initiation of breastfeeding (EIBF) and EBF in the first three days (EBF3D). It seems that social norms (in and outside the health facility) encourage CMF feeding and low self-efficacy towards breastfeeding, and these norms have spread widely across geographical regions and socioeconomic groups. Additional health systems practices delaying breastfeeding in the first days include medical procedures (high rates of cesarean sections and episiotomies), lack of skin-to-skin care (SSC), limited breastfeeding counseling, and widespread availability of CMF (either by mothers bringing it with them, purchasing at facilities, or receiving samples) [76]. The availability and marketing of CMF-PW in Viet Nam, including cross-promotion, have also likely contributed to its high use among pregnant women [64,65,76].

The full implementation of breastfeeding-supportive policies can contribute to a broader gender transformative agenda in both the Philippines and Viet Nam. This would include strategies such as increasing fathers’ involvement and support, societal shifts that encourage breastfeeding as the norm, improved individual counseling and group support programs, universal implementation of the BFHI/Mother–Baby-Friendly Initiative, and further support for vulnerable parents [76]. It is important that the economic contribution of unpaid care work, predominantly by women and including breastfeeding, is adequately recognized. The recently developed Mothers’ Milk Tool can be used to aid advocacy to acknowledge the economic and societal contributions of breastfeeding women, including in ensuring food security [3,4]. Given the vulnerability of both countries to climate change and natural hazards, breastfeeding advocacy should be integrated into climate and disaster response planning. The Green Feeding Tool demonstrates how investing in breastfeeding has the potential to mitigate the greenhouse gas impacts of CMF and highlights the importance of acknowledging breastfeeding in first-food systems advocacy [123]. Further research is needed to explore the alignment between health and labor-related policies to enable breastfeeding (such as the Code and maternity protection) with other social policies, such as childcare. However, scale-up of nutrition-sensitive policies that enable breastfeeding (such as national Code implementation and maternity protection) is essential to address structural barriers in a broad range of disciplines, including maternal, infant, and young child health and nutrition, gender equity in the labor sector, and women’s rights, and even has implications for climate. Another important consideration for future policy is the geographic disparities in breastfeeding practices documented in the Philippines and Viet Nam, which are projected to increase in the Philippines [72], indicating the need for geographic targeting of interventions.

The policy change processes in the Viet Nam have been supported by an initiative through efforts from - A&T, UNICEF, and partners in seven Southeast Asian countries, including Viet Nam using the Collective Impact framework [59]. This has expanded to other countries in the region and involves establishing partnerships, building evidence, developing messages and materials, and fostering consensus [59]. The process has been evaluated and challenges encountered during expansion, partly attributed to the complex nature of the initiative, provide insights into potential obstacles in harmonizing regulatory environments across the region. A key lesson was the need for flexibility and innovation at the regional level to provide backbone support to adapt when expanding to diverse country structures [59]. Interviews with policy stakeholders in Viet Nam emphasize the importance of broader policies promoting women and children’s rights for implementing national Code and maternity protection legislation [60]. Strong government relationships and coordination with non-governmental and international organizations facilitated policy revisions. However, effective implementation requires support from national and local governments.

ASEAN is a political and economic group of 10 countries in Southeast Asia including the Philippines and Viet Nam, representing over 600 million people. The ASEAN Health Cluster recently published minimum standards to facilitate a more harmonized regulatory environment in the region (for the Protection, Promotion and Support of Breastfeeding and Complementary Feeding, including Code implementation and Guidelines on Actions to Protect Children from the Harmful Impact of Marketing of Food and Non-alcoholic Beverages) [124,125]. The learnings from the Philippines and Viet Nam could provide a toolbox for other member states as they roll out these standards. Similar minimum standards for maternity protection could be developed to ensure regional policy coherence, leveraging member states’ commitments to ASEAN, as described in “The ASEAN Leaders Declaration on Ending All Forms of Malnutrition” of 2017 [126].

### 4.4. Feasibility of Recommendations

In this manuscript, we summarize key recommendations from previous studies. The feasibility of implementing these recommendations in the Code and maternity protection in the Philippines and Viet Nam depends on several factors and will require some adaptation of existing strategies, including responsiveness to opportunities and flexibility in approach. Addressing legal gaps and enhancing regulations restricting CMF promotion require strong political will and comprehensive legislative frameworks. Investing in social and behavior change communication strategies to raise awareness and promote breastfeeding involves significant resources and coordination among various stakeholders. Improving breastfeeding counseling and support necessitates training health professionals and engaging communities, which can be challenging but is essential for long-term success.

Increasing paid maternity leave and expanding coverage to the informal sector are ambitious but necessary steps for maternity protection. In the Philippines, institutionalizing maternity cash transfers for the informal sector and reducing contribution requirements are feasible with legislative support and adequate funding. In Viet Nam, expanding entitlements to a large portion of the labor force not currently covered by social insurance requires substantial policy changes and financial investment. Strengthening communication about maternity entitlements and ensuring supportive work environments are critical but require cultural shifts and employer buy-in. Effective monitoring and enforcement of policies, supported by inter-agency teams and integrated government monitoring, are crucial for ensuring compliance and protecting workers’ rights.

Although the recommendations are ambitious, they outline the next steps required for policy advocacy and program implementation. Previously in Viet Nam, successful advocacy for six months of paid maternity leave (2012) was motivated by children’s rights and the need for EBF for children under six months. Further successful workplace lactation support, which mandates 60 min paid breastfeeding breaks and lactation rooms in companies with over 1000 female employees (2021), was based on the argument that children need to continue breastfeeding until 24 months. These entitlements were considered not feasible at the time they were proposed yet were approved and implemented. While many recommendations require financial commitments, the cost of inaction is far greater than the investment needs, as illustrated by the cost of not breastfeeding estimates for both countries [3]. Strategies to enhance the financial feasibility of these recommendations could include supporting these countries to improve responsive public financial management, renewing financial commitments, and exploring innovative financing to support recommended actions [127]. There is an opportunity to deliver more for the resources available by improving the efficiency of spending. Untapped opportunities include leveraging universal health coverage financing and adaptive social protection and assistance programs, repurposing of subsidies, and accessing climate funds with ongoing efforts for breastfeeding investments to be recognized as carbon offsets [128].

### 4.5. Strengths and Limitations

This manuscript has been able to draw on extensive in-depth primary research using mixed methods conducted in the Philippines and Viet Nam. The primary research adapted existing questionnaires based on global standards. Much of the primary research was cross-sectional in nature and cannot be used to conclude on causal relationships, yet trends and associations have been described. We acknowledge that the selection of studies was purposeful and not systematic due to the scope of work we could undertake with the requirement of our available funding: this work involved using previously collected data or information from A&T, which may alter our understanding of the overall picture in the Philippines and Viet Nam. Still, since the synthesis was reviewed by authors involved in the primary data collection and familiar with the landscape, we anticipate that key literature has been included. We acknowledge that bias may exist due to the potential subjective nature of qualitative analysis but have attempted to reduce this through triangulation, reflexivity, and by checking interpretations with researchers involved in some of the primary data collection on which the synthesis is based.

## 5. Conclusions

The Philippines and Viet Nam have progressive policies to enable breastfeeding, associated with substantial increases in breastfeeding rates in recent years. Comprehensive implementation research has documented policy progress, remaining barriers, and improvement in breastfeeding rates in these two countries. We have been able to collate the challenges identified and lessons learned from successful policy advances in breastfeeding support in these two countries, which could serve as valuable insights when advocating for improved policy to enable breastfeeding in other countries in the region and potentially other LMICs. The applicability of the lessons learned from these two countries would depend on knowledge, attitudes, and barriers specific to other contexts. Learnings from the implementation of stronger national marketing restrictions of CMF and associated products, including adequate monitoring and enforcement, and maternity protection policy aligned with global standards in the Philippines and Viet Nam can inform the implementation of newly developed regional standards together with supportive or related policies, facilitating the harmonization of regulatory environments in the region.

## Figures and Tables

**Figure 1 healthcare-13-00544-f001:**
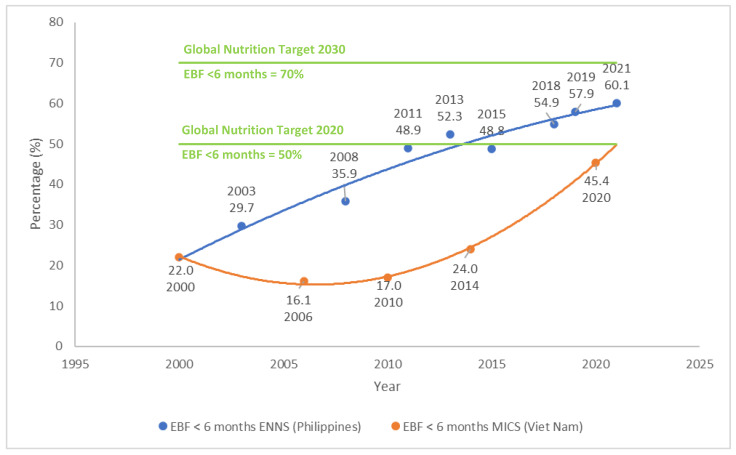
Exclusive breastfeeding under 6 months in the Philippines and Viet Nam. Abbreviations: EBF: Exclusive breastfeeding; ENNS: Expanded National Nutrition Survey; MICS: Multiple Indicator Cluster Survey. Sources of data: Philippines Expanded National Nutrition Survey (ENNS) of 2019 and 2021 [70,71]; UNICEF Data Warehouse: Viet Nam Exclusive breastfeeding (0–5 months) [73].

**Figure 2 healthcare-13-00544-f002:**
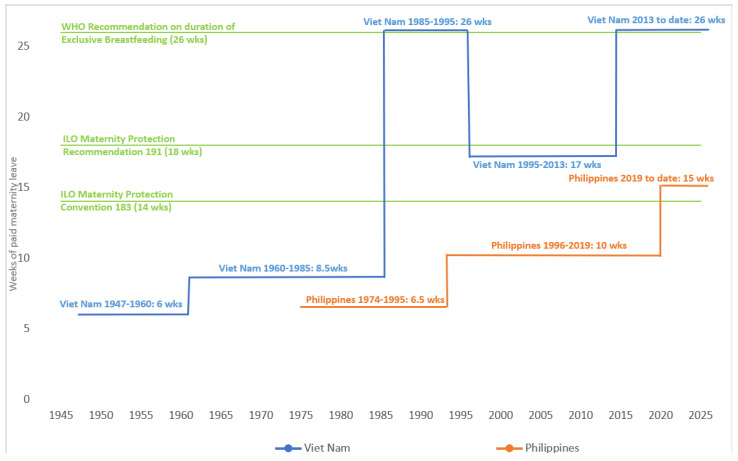
Timeline of changes in paid maternity leave provisions in the Philippines and Viet Nam from 1947 to present [66,69]. Abbreviations: EBF: Exclusive breastfeeding. ILO: International Labour Organization; MPC: Maternity Protection Convention; MPR: Maternity Protection Recommendation; WHO: World Health Organization.

**Table 1 healthcare-13-00544-t001:** Published research assessing the effectiveness of policies relating to breastfeeding protection, promotion, and support in the Philippines and Viet Nam.

Title	Country, Year of Data Collection	Study Design	Study Sample, Data Collection Techniques, and Data Used	Aim	Policy Type Investigated
Translating the International Code of Marketing of Breast-milk Substitutes into national measures in nine countries [63]	Viet Nam, May 2015 to March 2017	Real-time evaluation	Participant observation 16 key informant meetings 3 in-depth interviews (IDIs) Reflective practice; desk review	To document the extent to which policy objectives were (or were not) achieved in 9 countries (including Viet Nam) and to identify the key drivers of policy changes	Breastfeeding protection: the Code
Implementation of the Code of Marketing of Breast-milk Substitutes in Vietnam: marketing practices by the industry and perceptions of caregivers and health workers [64]	Viet Nam, May to July 2020	Mixed methods, cross-sectional	Quantitative survey of 268 pregnant women and 726 mothers of infants aged 0–11 months Qualitative IDIs with 70 participants, incl. subsets of interviewed women (n = 39), policymakers, media executives, and health workers (n = 31)	To examine the enactment and implementation of the Code of Marketing of Breast-milk Substitutes (the Code) in Viet Nam, focusing on marketing practices by the baby food industry and perceptions of caregivers, health workers, and policymakers	Breastfeeding protection: the Code
Beliefs and norms associated with the use of ultra-processed commercial milk formulas for pregnant women in Vietnam [65]	Viet Nam, May to July 2020	Post hoc analysis of quantitative survey data	Quantitative interviews with 268 pregnant women in their second and third trimesters from two provinces and one municipality representing diverse communities in Viet Nam	To examine the association between the use of commercial milk formula for pregnant women and related beliefs and norms among pregnant women in Viet Nam	Breastfeeding protection: the Code
Awareness, perceptions, gaps, and uptake of maternity protection among formally employed women in Vietnam [66]	Viet Nam, May to July 2020	Mixed methods, cross-sectional	Quantitative interviews with 494 formally employed female workers (107 pregnant and 387 mothers of infants) IDIs with a subset of women (n = 39)	To examine the uptake of Viet Nam’s maternity protection policy in terms of entitlements and awareness, perceptions, and gaps in implementation through the lens of formally employed women	Maternity protection
Implementation and effectiveness of policies adopted to enable breastfeeding in the Philippines are limited by structural and individual barriers [67]	The Philippines, December 2020 to March 2021	Mixed methods, cross-sectional	Desk review of policies and documents IDIs with 100 caregivers, employees, employers, health workers, and policymakers in the Greater Manila Area	This study assesses the adequacy and potential impact of breastfeeding policies, as well as the perceptions of stakeholders of their effectiveness and how to address implementation barriers	Breastfeeding protection promotion and support, including the Code and maternity protection
The impact of Vietnam’s 2013 extension of paid maternity leave on women’s labour force participation [68]	Viet Nam, 2015–2018 (data from Labor Force Surveys)	Regression discontinuity (RD) design	RD to evaluate the impact of paid maternity leave on the probability of women holding a job and formal labor contract 3–5 years after giving birth	To evaluate whether the expansion of Viet Nam’s paid maternity leave policy was associated with improved long-term labor outcomes for Vietnamese women	Maternity protection
Maternity protection policies and the enabling environment for breastfeeding in the Philippines: a qualitative study [69]	The Philippines, December 2020 to April 2021	Mixed methods, cross-sectional	Desk review of policies, guidelines, and related documents on maternity protection. IDIs with 87 mothers and partners, employers, and authorities from government and non-government organizations in the Greater Manila Area	This study reviewed the content and implementation of maternity protection policies in the Philippines, assessed their role in enabling recommended breastfeeding practices, and identified bottlenecks to successful implementation	Maternity protection

**Table 2 healthcare-13-00544-t002:** Sociodemographic indicators for the Philippines and Viet Nam.

Sociodemographic Indicator	Philippines	Viet Nam
Population	115,559,009 [77]	98,186,856 [78]
Urban: rural	48:52 [79]	39:61 [79]
Life expectancy	69.3 years [80]	73.6 years [80]
Fertility rate	1.9 (2022) [81]	1.9 [80]
Institutional birth rate	89% (2020) from 50.5% (2010)	96.3%
Crude birth rate (per 1000 people)	21.8 (2021) [82]	15.0 (2021) [82]
Public: private hospitals	40:60	86:14
Exclusive breastfeeding < 6 mo.	34.0% (2008); 60.1% (2021) [71]	17.0% (2010); 45.4% (2021) [83]
Continued breastfeeding (% children 12–23 months fed breastmilk the previous day)	57.1% (2022) [81]	43.9% (2020) [83]
Immediate skin-to-skin contact	71% [81]	59% [84]
Stunting in children < 5 years	26.7% (2021) [71]	19.5%
Unemployment (2023)	4.4%	2.0%
Labor force participation rate	Men: 76.3%; women: 52.9% [85]	Men: 74.3% Women: 61.6% [86]
Vulnerable employment ^1^	2022: Men: 30%; women: 38.5% [85]	2022: Men: 46.9%; women: 57.3% [48]
Informal employment	38.9% [46]	2019: Men: 78.9%; women: 67.2% [49]
Commercial milk formula market	2020: 8th largest globally: USD 832.2 million in total; USD 7.6 annual per capita expenditure [87]	2020: 4th largest globally: USD 1421.2 million in total; USD 14.6 annual per capita expenditure [87]

^1^ “Workers in vulnerable employment are the least likely to have formal work arrangements, social protection, and safety nets to guard against economic shocks; thus, they are more likely to fall into poverty.”—https://genderdata.worldbank.org/countries/philippines (accessed on 16 February 2025).

**Table 3 healthcare-13-00544-t003:** Implementation status of the Code in the Philippines and Viet Nam: Scores allocated per provision extracted from the 2024 Code Status Report [17].

Provision	Philippines	Viet Nam	Highest Scores
Scope	20	16	20
Monitoring and enforcement	10	10	10
Informational/educational materials	9	5	10
Promotion to general public	10	20	20
Promotion in health facilities	10	10	10
Engagement with health workers and systems	14	10	15
Labeling	12	8	15
Total	85	79	100

**Table 4 healthcare-13-00544-t004:** Summary of maternity protection provisions and paternity leave in legislation in the Philippines and Viet Nam (extracted from 2022 ILO Care at work report) [41].

	ILO Maternity Protection Convention 183	ILO Maternity Protection Recommendation 183	Philippines	Viet Nam
Paid maternity leave				
Duration of maternity leave in national legislation	Mandates minimum maternity leave of 14 weeks	Recommends increasing maternity leave to 18 weeks	15 weeks (105 days), option to extend by an additional 30 days	26 weeks (6 months)
Amount of maternity leave cash payments (% of previous earnings)	Adequate to keep mother and child healthy, out of poverty, especially women in informal economy; >67% of previous earnings	Recommends increasing maternity leave cash payments to 100%, when possible	100% for 15 weeks (105 days)	100%
Source of funding maternity leave cash payments	Employers should not be individually liable for direct costs of maternity leave. Cash benefits shall be provided through compulsory social insurance, public funds, or non-contributory social assistance to women who do not qualify for benefits out of social insurance; especially for informal economy or self-employed workers		Social insurance and employer	Social insurance only
Maternity leave cash payments for self-employed workers			Yes, but only for workers who are actively paying members of the Social Security System	No
Paternity leave			7 days	5 days
Source of funding			Employer	Social insurance
Breastfeeding (nursing) breaks				
Entitlement to paid nursing breaks	Women should be provided with the right to one or more daily breaks or daily reduction of work hours to breastfeed. The period during which this is allowed, the number and duration of breaks, and procedures for reducing daily work hours shall be determined by national law	Frequency and length of nursing breaks should be adapted to needs. It should be possible to combine time allotted for daily nursing breaks to allow reduced work hours at beginning/end of the workday. Where practical, provision should be made for establishing hygienic nursing facilities at or near the workplace	Paid	Paid
Number of daily nursing breaks			Not limited	Not specified
Total daily nursing break duration			40 min	60 min
Period when nursing breaks are allowed by law			Not specified	Until child is 12 months
Statutory provisions for working nursing facilities			All workers	Mandatory at workplaces ≥ 1000 female employees

**Table 5 healthcare-13-00544-t005:** Key barriers to implementing legislation to enable breastfeeding in the Philippines and Viet Nam.

The Philippines	Viet Nam
1. Barriers to implementing Code legislation	
Structural gaps in legislation allowing promotion to the general public and insufficient labeling provisions [17,67]. Inadequate restrictions on industry-funded research and sponsorship of health professionals and academics, resulting in conflicts of interest (COIs) [96]. Even though the Philippines scores 10/10 for monitoring and enforcement in the global Code Status Report, weak monitoring and inadequate enforcement of the Philippine Milk Code have been recently documented from interviews with health workers and policymakers [67]. DoH has primary responsibility for implementing the Philippines Milk Code but has still not established the monitoring teams mandated in the 2006 Rules and Implementing Regulations (IRR) and does not conduct regular inspections [96]. Ambiguity surrounding monitoring responsibilities and irregular inspections hamper enforcement [67]. Weak sanctions limit enforcement. Some consider existing legislation to be too strict, especially the prohibition on product donations during emergencies [67].	Gaps in legislation: insufficient scope, inadequate regulation of information and education materials, engagement with health workers and systems, and labeling, illustrated by continued violations [64]. Conflicting advertising regulations regarding functional food, supplemented food, food for special medical purposes for children under 24 months, and breastmilk substitutes. Legislative gaps mean company representatives still access health facilities and obtain contact information from pregnant women and new mothers [64]. Gaps in scope allow rampant cross-promotion, especially of CMF-PW with CMF for infants. Companies target pregnant women for CMF-PW using tactics otherwise prohibited. Routine monitoring is limited, relies on self-assessment (through hospital accreditation, where the Code is one of 83 quality standards), and data is unavailable. Limited enforcement due to human resource constraints and pro-industry tendencies. Industry-sponsored research has influenced national nutrition guidelines for pregnant and lactating women. Industry also sponsors health professionals’ attendance at events and provides financial support to health centers, creating conflicts of interest and leading to product promotion by health professionals [65]. No specified agency to monitor digital marketing, which CMF manufacturers exploit [96].
2. Barriers to implementing maternity protection	
Informal sector workers are not reached by maternity protection entitlements and the Social Security System (SSS) only reaches 54% of workers nationally [69]. Length of paid maternity leave less than WHO recommendation of EBF to six months. Employers recognize the value of maternity protection but perceive disadvantages to policies with some not supporting workplace lactation, with policies varying according to workplace and type of work [69]. Few mothers use available facilities due to perceived inconvenience and challenges such as lack of equipment, workload, inadequate breaks, and unsuitable environments, particularly in the informal sector. Women in output-based jobs face the difficult choice between breastfeeding and income generation [69]. Some employers exploit short-term contracts to avoid maternity entitlements. Sense of acceptance that breastfeeding will stop when women return to work. No systematic enforcement and monitoring, unclear roles for government agencies, and limited workplace inspections further hinder effective implementation [69]. Voluntary mechanisms like the Mother–Baby-Friendly Workplace Certification lack sufficient uptake.	Low knowledge among all mothers of the full set of maternity protection entitlements. Limited awareness of the range of benefits that paid maternity leave is associated with. Perceived barriers to using entitlements. Disparities in knowledge and uptake by occupation and sector. Limited access to cash entitlements while on maternity leave and low maternity allowance do not protect mothers and infants from poverty due to low contribution to social security fund before maternity leave. Discrimination based on pregnancy and childbirth continues. Some women mistakenly perceived maternity protection as employer-provided benefits rather than legal entitlements financed through social insurance [66]. Unintended negative consequences on labor force participation [68].

CMF-PW: Commercial milk formula for pregnant women.

**Table 6 healthcare-13-00544-t006:** Summary of recommendations to improve implementation of policies to enable breastfeeding.

Recommendations to Improve Implementation of Code Legislation in the Philippines and Viet Nam
Address remaining legislative gaps: ○ ***Philippines:*** Stricter regulation on “promotion to the general public”; clear definitions; restrict sponsorship of health workers and research by the CMF industry to prevent conflicts of interest. ○ ***Viet Nam:*** Better regulation of CMF-PW promotion; extend restrictions to CMF for breastfeeding women; stricter provisions on CMF promotion through health facilities, public, social media, and industry sponsorship; communicate harmful effects of using CMF-PW to women and health workers. ○ ***Both countries:*** Improve coverage and regulate digital marketing of CMF, aligned with WHO guidance. Implement a social and behavior change communication strategy to raise awareness of the legislation, including the scope and need for marketing prohibitions, and promote breastfeeding, addressing common misconceptions. Improve breastfeeding counseling for mothers and family members during pregnancy, at birth, and in the early days after birth: Breastfeeding information should be combined with skilled peer counseling and psychosocial support to enhance self-efficacy. Communities, family members, peers, coworkers, employers, and health systems (i.e., whole of society) need to be engaged due to their influence on individual knowledge, attitudes, and practices. Improved coverage and quality of breastfeeding support: Immediate and prolonged skin-to-skin contact, non-separation of the mother and baby, and reducing unnecessary cesarean section births; increase MBFHI coverage; IYCF support groups (at health facilities, in communities, online); comprehensive workplace lactation support. Continuous capacity building of health professionals, pre-/post-service, including the need for Code legislation. Institutionalized ongoing monitoring (using eight steps suggested in NetCode protocol for ongoing monitoring): (a) **Negotiate the political and bureaucratic environment:** Improved political commitment and leadership to strengthen breastfeeding protection and local financing of breastfeeding promotion. (b) **Determine coverage and extent of monitoring:** Including non-hospital settings (clinics, private sector); mobilizing community monitoring, using new technologies (e.g., AI) for monitoring and enforcing legislation. (c) **Build a national monitoring team:** Engage mothers and the public in monitoring the Code. (d) **Cost and budget of monitoring.** (e) **Develop standard monitoring tools and a database** especially at local levels (e.g., through local regulation). The Centers of Excellence for Breastfeeding in Viet Nam are a good model for health facilities. (f) **Build the capacity of monitors:** Continuous training and professional development. (g) **Monitor and enforce**: Rigorous enforcement, including prosecution of violators. (h) **Evaluate and assess:** Monitoring and enforcement activities (such as health inspection visits) should be compiled, analyzed, and published, and/or there should be a database available to track this [64,65,67,96,99].
**Recommendations to Improve Implementation of Maternity Protection Legislation in the Philippines and Viet Nam**
***The Philippines:*** Increase paid maternity leave to 6 months. In both countries, improve coverage and accessibility of paid maternity leave, especially in the informal sector. ○ ***The Philippines:*** Maternity cash transfers for the informal sector are affordable [43], institutionalization through legislation is being explored; contributions required by informal sector workers should decrease. ○ ***Viet Nam:*** Around 2/3 of the labor force is ineligible for paid maternity leave, entitlements need to expand to the informal sector, those who have not contributed to the social insurance fund long enough to receive entitlements, or to all women, regardless of employment status. Strengthen communication about maternity entitlements for mothers and obligations of employers. ○ Breastfeeding promotion and support at workplaces beyond complying with minimum requirements. ○ Supportive work environments should encourage women to use maternity protection entitlements. ○ Improve awareness of maternity entitlements and social security programs. ○ Many workers perceive maternity protection as a detriment to employers, there is a need to shift this perspective. ○ Ensure that policymakers, employers, and employees are aware of the latest labor laws and recent changes. Improve monitoring and enforcement (M&E) of existing policies: ○ Inter-agency monitoring teams for all levels, as mandated by law. ○ Integrate M&E in routine monitoring of other government programs. ○ Improve identifying and penalizing employers who do not comply with regulations. Paid leave should be fully financed through social insurance or public funding, including in the private sector. Partnerships between government and workplaces should ensure continuity of advocacy for breastfeeding programs and provide technical assistance to promote breastfeeding in workplaces. Integrate maternity protection policies with breastfeeding promotion programs. Strengthen and expand the Mother–Baby-Friendly Workplace Certification. ***Viet Nam***: A streamlined digital system needed for submitting documents and transferring funds online during maternity leave; review whether returning to work after four months with doctor’s approval is beneficial; increase female employees’ awareness of their right to negotiate additional unpaid leave; improve monitoring of employer compliance with local regulations; expand mandatory workplace lactation regulations to all companies [66,69].

## Data Availability

All data generated or analyzed during this study are included in this article and/or appropriately cited to publicly accessible resources.

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
