# Peer review of "Evidence-Based Lessons from Policy Implementation Research in Two Countries Achieving Progress on Global Breastfeeding Targets: Recommendations from the Philippines and Viet Nam"

_healthcare, 2025, doi:10.3390/healthcare13050544_

Round 1

Reviewer 1 Report

Comments and Suggestions for Authors

I have read this paper with great interest, and with a background on perinatal clinical research, including breastfeeding practices. There is for sure value in this paper, describing the impact of policy support on breastfeeding practices. These policies are subsequently described in a structured as evidence synthesis. I’m hereby very impressed by eg table 1 on how well different aspects of this implementation have been reported in different types of study designs.

While there is value in the paper, the underlying assumption is that the same approach will either result in further progress in these countries, or would be as effective in other coutries, while this very likely also depend on knowledge, attitudes and barriers specific to another region. I rather perceive this paper as providing a ‘toobox’, as interventions that were effective in a specific setting, and therefore potentially also effective elsewhere.

I would like to encourage the authors to further reflect on next steps to be taken. Do you ‘believe’ that ‘more of the same’ will be effective to further improve the setting, or are other type of interventions warranted to support practices.

Are there contemporary qualitative type of data on why individuals opt for breastfeeding (initiation, continuation) or formula ? It is not impossible that – to further improve the setting in both countries – a more targeted or personalized approach in ‘underperforming groups of postpartum women’ is warranted. This comment might be affected by a paper (under revision) on the maternal socioeconomic status and how these factors have a different impact on breastfeeding initiation or duration in different regions of the world (in essence, over one decade, practiced doubled).

Author Response

Thank you very much for taking the time to review this manuscript. Please find the detailed responses below and the corresponding revisions highlighted in red text in the re-submitted files.

Comments 1: I have read this paper with great interest, and with a background on perinatal clinical research, including breastfeeding practices. There is for sure value in this paper, describing the impact of policy support on breastfeeding practices. These policies are subsequently described in a structured as evidence synthesis. I’m hereby very impressed by eg table 1 on how well different aspects of this implementation have been reported in different types of study designs.
While there is value in the paper, the underlying assumption is that the same approach will either result in further progress in these countries, or would be as effective in other countries, while this very likely also depend on knowledge, attitudes and barriers specific to another region. I rather perceive this paper as providing a ‘toolbox’, as interventions that were effective in a specific setting, and therefore potentially also effective elsewhere.

Response 1: Thank you for pointing this out. We agree with this comment. Therefore, we have made the following edits:
-    added the word ‘potentially’ to the abstract (Pg 1, line 34-35) and conclusions statement (Pg 21, line 646-647) when describing how the lessons learned could be applied. 
-    added that the learnings from these 2 countries could provide a toolbox for other countries, in the Discussion, we have (Pg 19-20, Lines 579-580). 
-    added the following sentence to the conclusion (Pg 21, lines 647-648): “The applicability of the lessons learned from these two countries would depend on knowledge, attitudes and barriers specific to other contexts.” 
-    We have made it clearer about the scope of the study in the Methods and acknowledge study limitations in the discussion.

Comments 2: I would like to encourage the authors to further reflect on next steps to be taken. Do you ‘believe’ that ‘more of the same’ will be effective to further improve the setting, or are other type of interventions warranted to support practices.

Response 2: Thank you for this comment. We have done some reflection on the next steps, in Table 6 of the results (Pg 15), and we also have some discussion about this, in sub-section 4.4 (feasibility of the recommendations) (Pg 16, lines 587-589). 

Comments 3: Are there contemporary qualitative type of data on why individuals opt for breastfeeding (initiation, continuation) or formula? It is not impossible that – to further improve the setting in both countries – a more targeted or personalized approach in ‘underperforming groups of postpartum women’ is warranted. This comment might be affected by a paper (under revision) on the maternal socioeconomic status and how these factors have a different impact on breastfeeding initiation or duration in different regions of the world (in essence, over one decade, practiced doubled).

Response 3: Thank you for this feedback. We look forward to seeing the paper described by the reviewer. To address the reviewer feedback, we have added a paragraph to the end of section 4.1 in the discussion about the Code, stating: “There may be many factors influencing parents and caregivers’ infant feeding practices, and these influences may have different impacts on breastfeeding initiation and continuation in different regions of the world. However, it is important that choices relating to infant feeding are made in an enabling environment that is free from commercial influence. The value of the Code is that it aims to protect all infants from harmful marketing, whether they are fed with breastmilk or infant formula.” (Pg 17, Lines 447-452). 

4. Response to Comments on the Quality of English Language N/A

5. Additional clarifications N/A

Reviewer 2 Report

Comments and Suggestions for Authors

The Pereira-Kotze et al article explores the policy and guidelines implemented in two cultural settings, Philippines and Viet Nam, to improve breastfeeding and protect maternity. The authors deeply described the evidence in sociocultural, demographic and economic characteristics of both contexts to extract insights, programs, gaps and actions to improve breastfeeding practices. Although the effort is huge, the manuscript is a bit disoriented, some sections too much extensive, losing the focus and doing difficult the readability. The suggestions would be:

Introduction. It is a confuse section, it is not clear what would be the propose it. For example, at the beginning, the authors stablished the promote breastfeeding, mention many organizations and guidelines, then, continue with comprehensive maternity protection and fulfillment women´s reproductive rights. Subsequently, in line 116, it starts to mention Philippines and Viet Nam, but the authors described CRC guidelines, which is out of standing. In line 141-142 the authors defined the research question. However, the proposal of this article (lines 70-72) is not merge with the aims described (lines 149-152).

I strongly recommend summary this section and deeply focus on what would be the main message to transmit. Additionally, it is not clear if the authors perform a systematic review, literature review, or what type of the article would be.

Material and methos. It is not clear what the type of the manuscript is, and it is difficult to understand the methodology strategy. The methods would not be replicable because it is not clear how the documents were kept. In addition, the table 1 looks like a result rather than methodology.

·       What does the Alive & Thrive initiative mean? Is a method to extract data? To search the references? To analysis the data? Please clarify.

·       Many of the info stablished in section 2.2 could be implemented in the introduction section, but summarized, because is the rationale of setting population explored. In addition, table 2 and figure 1 is not need due to data are reported in the main text (or maybe upload it as a supplementary material).  

Results. This section is extensive, and the focus is lost. Sometimes, the results and discussion are mixed, particularly in sections 3.4.1. and 3.4.2. For example, sections 3.2 and 3.3 could be the main interest of the article, but the way to described is confused, being “the expanded Breastfeeding Promotion Act of 2009 mandates” the nice part described.

·       How do the data form table 3 were obtained? In addition, the figure 2 is not clear, what does represent? The lines labels do not merge with the axis-x.

·       The table 5 is redundant with the text described. Please, summarize it.

Discussion. In many cases, data from results section are repeated (for example, line 526-528, or line 540-542). It does not feel like a discussion, where evidence is compared or controversial argued. In addition, what would be the main funding? This need to be places at the beginning of the section. Furthermore, were the aims fulfillment? Also need to be clary in the continuation. It would be strongly recommended that authors summarized this section, focus on the main message, giving fluency to the text.

·       Line 551. How the colostrum powder could intake until six months?

·    Line 769. Qualitive analysis involves triangulation and report themes and subthemes also describing the strategy of analysis of the documents. Please, clarify.

Conclusions. It is clear the strong effort performed by these two socio-cultural setting to improve EBF strategies. I agree with the authors that its implementations could help to other settings, but the conclusion is round circles in this idea, without to stablish what kind of policy to implement or to avoid the barriers.

Minor comments

·       Lines 76-78 is difficult to understand.

·       Table 1. First column does not describe author and year.

·       Line 241. What does mean read and re-read?

·       Line 253-255. It is confused the way of sentence was written.

·   Describe directly the real number, for example, line 371 describe 91.7% rather than “More than 90%”.

·       Line 501 and 761. Write “manuscripts”, “documents” or “articles” rather than “papers”.

·  There are many acronyms, please consider to delete it or implement abbreviations list.

Comments on the Quality of English Language

The sections are unclear, some sentences are repeted and ideas are no organized.

Author Response

Thank you for taking the time to review this manuscript. Please find the detailed responses below and the corresponding revisions highlighted in red text in the re-submitted file.

Comments 1: The Pereira-Kotze et al article explores the policy and guidelines implemented in two cultural settings, Philippines and Viet Nam, to improve breastfeeding and protect maternity. The authors deeply described the evidence in sociocultural, demographic and economic characteristics of both contexts to extract insights, programs, gaps and actions to improve breastfeeding practices. Although the effort is huge, the manuscript is a bit disoriented, some sections too much extensive, losing the focus and doing difficult the readability. The suggestions would be:

Introduction. It is a confuse section, it is not clear what would be the propose it. For example, at the beginning, the authors stablished the promote breastfeeding, mention many organizations and guidelines, then, continue with comprehensive maternity protection and fulfillment women´s reproductive rights. Subsequently, in line 116, it starts to mention Philippines and Viet Nam, but the authors described CRC guidelines, which is out of standing. In line 141-142 the authors defined the research question. However, the proposal of this article (lines 70-72) is not merge with the aims described (lines 149-152).
I strongly recommend summary this section and deeply focus on what would be the main message to transmit. Additionally, it is not clear if the authors perform a systematic review, literature review, or what type of the article would be.

Response 1: Thank you for the feedback. 
-    We have moved quite a lot of content about the Philippines and Viet Nam, from the Study setting section of the Methods to the Introduction section (Pg 2, Lines 76-92). 
-    We have removed some sentences and details from the text that might have been unnecessary or duplicated, and have tried to summarise text. 
-    We have put all the information relevant to the CRC in one paragraph, and hope that the way it is now presented, makes better sense (Pg 3, lines 136-144). 
-    We have made edits to align the wording of the research question and aim of the study (in the Abstract – Pg 1, lines 20-22). 
-    We have clarified in section 2.1 study design, that the methods are “a qualitative evidence synthesis” using purposive sampling (Pg 4, line 171). 

Comments 2: Material and methods. It is not clear what the type of the manuscript is, and it is difficult to understand the methodology strategy. The methods would not be replicable because it is not clear how the documents were kept. In addition, the table 1 looks like a result rather than methodology.
·       What does the Alive & Thrive initiative mean? Is a method to extract data? To search the references? To analysis the data? Please clarify.
·       Many of the info stablished in section 2.2 could be implemented in the introduction section, but summarized, because is the rationale of setting population explored. In addition, table 2 and figure 1 is not need due to data are reported in the main text (or maybe upload it as a supplementary material).  

Response 2: Thank you for the feedback and helpful suggestions, we have made the following changes to improve the clarity and reduce confusion. 
-    In section 2.1 (study design) we describe that this manuscript is “an evidence synthesis of seven purposively selected primary research publications from the Philippines and Viet Nam”, indicating that study selection was purposeful. We have added the word ‘qualitative’ to describe that this was a qualitative evidence synthesis. We have added another sentence to this paragraph to provide more clarity: “The original studies were selected, because they were conducted in the Philippines and Viet Nam, were coordinated by A&T, and their scope included the Code and/or maternity protection (i.e., strategies to protect breastfeeding).” (Pg 4, Lines 171-183).
-    We have provided a description of the A&T initiative together with a reference to support this (Pg 4, Line 175-180).
-    Figure 2 provides a clear graphic illustration that is helpful for readers, and we would like to keep it in as it provides a good visual. 
-    We have left Table 2 in the manuscript, but rather moved quite a lot of the text and words, to reduce repetition of data presented in the table. 
-    We have moved quite a bit of content from the Study setting section (2.2) to rather appear in the Introduction, and have tried to reduce the total amount of content. 

Comments 3: Results. This section is extensive, and the focus is lost. Sometimes, the results and discussion are mixed, particularly in sections 3.4.1. and 3.4.2. For example, sections 3.2 and 3.3 could be the main interest of the article, but the way to described is confused, being “the expanded Breastfeeding Promotion Act of 2009 mandates” the nice part described.
·       How do the data form table 3 were obtained? In addition, the figure 2 is not clear, what does represent? The lines labels do not merge with the axis-x.
·       The table 5 is redundant with the text described. Please, summarize it.

Response 3: We agree with the feedback.
-    Thank you, we have removed sections 3.4.1 and 3.4.2, and rather kept table 5 (adding a little detail) – and to avoid duplication. 
-    For Table 3, we have provided the reference and indicated that the data was extracted from the Code Status Report (Pg 9, line 251).
-    For figure 2, we have edited the caption and reformatted the table, to align the line labels to appear appropriately with the correct date on the x-axis (Pg 9, lines 255-257). Thank you for the comment.

Comments 4: Discussion. In many cases, data from results section are repeated (for example, line 526-528, or line 540-542). It does not feel like a discussion, where evidence is compared or controversial argued. In addition, what would be the main funding? This need to be places at the beginning of the section. Furthermore, were the aims fulfillment? 

Also need to be clary in the continuation. It would be strongly recommended that authors summarized this section, focus on the main message, giving fluency to the text.
·       Line 551. How the colostrum powder could intake until six months?
·    Line 769. Qualitive analysis involves triangulation and report themes and subthemes also describing the strategy of analysis of the documents. Please, clarify.

Response 4: Thank you for this feedback.
-    We have added a paragraph to the beginning of the discussion to summarise the main findings and whether the aims were fulfilled (Pg 16, lines 383-396). 
-    On Pg 17-18, Lines 466-490 discuss some of the conflicting findings, Lines 502-510 describe recent literature.
-    We have removed content from the discussion that was duplicated from the results.  
-    We have edited the wording around colostrum powder to clarify the meaning (Pg 16, line 415-416). 
-    We have added that triangulation was used in the analysis (Pg 21, line 636). 

Comments 5: Conclusions. It is clear the strong effort performed by these two socio-cultural setting to improve EBF strategies. I agree with the authors that its implementations could help to other settings, but the conclusion is round circles in this idea, without to stablish what kind of policy to implement or to avoid the barriers.

Response 5: Thank you for the feedback. In section 3.5 of the Results, there are examples of potential policy changes that could improve the environment for breastfeeding and reduce barriers to implementation in the two countries (synthesized from the 7 studies included). More detailed descriptions of our recommendations can be found in Table 6, as well as in the corresponding text in the results and discussion section 4.4. We have added an introductory paragraph to the discussion section, and also added some clarifying wording to the conclusion paragraph. We hope that this provides clarity regarding the kind of policies that need to be implemented. 

Comments 6: Minor comments
· Lines 76-78 is difficult to understand.
· Table 1. First column does not describe author and year.
· Line 241. What does mean read and re-read?
· Line 253-255. It is confused the way of sentence was written.
· Describe directly the real number, for example, line 371 describe 91.7% rather than “More than 90%”.
· Line 501 and 761. Write “manuscripts”, “documents” or “articles” rather than “papers”.
· There are many acronyms, please consider to delete it or implement abbreviations list.

Response 6: Thank you for the minor comments feedback, we have incorporated all of these accordingly: 
•    We have reworded what was previously lines 76-78 (now, Pg 2, lines 64-65).
•    Thank you for identifying the error in the title of the first column of Table 1, we have removed the words ‘author’ and ‘year’. (Pg 5-6)
•    Removed the word “re-read” from line 218 (Pg 8). 
•    We have improved the wording in what was previously lines 253-255 (now, Pg 8, lines 231-232).
•    In line 371, we have removed the text stating “more than 90%” (now, Pg 13, line 344). 
•    In line 360 (Pg 14), we have replaced the word ‘papers’ with ‘articles’ and in line 585 (pg 20), we have replaced the word ‘paper’ with ‘manuscript’.
•    There is an abbreviations list, but the formatting of the journal requires that the abbreviations list is at the end of the paper (just before the references section, Pg 22, 675-674).

4. Response to Comments on the Quality of English Language
Point 1: The sections are unclear, some sentences are repeted and ideas are no organized.
Response 1: We have tried to improve the wording throughout the manuscript and removed repetition. 

5. Additional clarifications N/A

Reviewer 3 Report

Comments and Suggestions for Authors

The changes in national Code implementations and exclusive breastfeeding (EBF) rates across two countries have been beautifully synthesized. I commend the researchers for their excellent work.

Lines 56–69: There is no need to include the five policy actions of the Global Breastfeeding Collective in detail, as these are already described in the text and are widely known.

It is recommended to reduce repetitions in the manuscript. For instance, exclusive breastfeeding rates are mentioned multiple times. Additionally, the findings presented in the text and the table are redundant and can be streamlined for brevity.

The referencing format should follow a consistent standard. All figures, tables, and boxes (e.g., Box 1, Table 3, Figure 2) should include proper citations where applicable.

Author Response

Thank you very much for taking the time to review this manuscript. Please find the detailed responses below and the corresponding revisions highlighted in red text in the re-submitted files.

Comments 1: The changes in national Code implementations and exclusive breastfeeding (EBF) rates across two countries have been beautifully synthesized. I commend the researchers for their excellent work.
Lines 56–69: There is no need to include the five policy actions of the Global Breastfeeding Collective in detail, as these are already described in the text and are widely known.

Response 1: Thank you for pointing this out. We agree with this comment, and it is a helpful way to reduce some of the word count. Therefore, we have removed the bulleted list (that previously appeared on Pg 2) and rather replaced with a much shorter summary sentence (Pg 2, lines 55-59). 

Comments 2: It is recommended to reduce repetitions in the manuscript. For instance, exclusive breastfeeding rates are mentioned multiple times. Additionally, the findings presented in the text and the table are redundant and can be streamlined for brevity.

Response 2: Thank you for the comment, we agree that unnecessary repetitions should be removed and have tried to do this throughout the manuscript through a thorough editing process. At the moment, the EBF rates are mentioned in the Abstract and in the study setting section of the methods (with reference to a figure) and we feel that this is appropriate for EBF rates to be mentioned in both places. We have tried to reduce repetitions elsewhere. 

Comments 3: The referencing format should follow a consistent standard. All figures, tables, and boxes (e.g., Box 1, Table 3, Figure 2) should include proper citations where applicable.

Response 3: Thank you for the feedback, we agree. We have, accordingly, revised:
-    For Box 1, we have included citations to all documents.
-    For Table 3, we have changed the reference to be a citation.
-    For Figure 2, we have added 2 citations to support the information in the figure.
-    For Table 4, we have added an appropriate citation to support the information in the table.

4. Response to Comments on the Quality of English Language N/A

5. Additional clarifications N/A

Round 2

Reviewer 2 Report

Comments and Suggestions for Authors

Thank you for this reviewed version of the manuscript.